# The Role of Mitochondrial Calcium Homeostasis in Alzheimer’s and Related Diseases

**DOI:** 10.3390/ijms21239153

**Published:** 2020-12-01

**Authors:** Kerry C. Ryan, Zahra Ashkavand, Kenneth R. Norman

**Affiliations:** Department of Regenerative and Cancer Cell Biology, Albany Medical College, Albany, NY 12208, USA; ryank8@amc.edu (K.C.R.); ashkavz@amc.edu (Z.A.)

**Keywords:** mitochondria, calcium, neurodegeneration, Alzheimer’s disease, MCU, ROS, presenilin

## Abstract

Calcium signaling is essential for neuronal function, and its dysregulation has been implicated across neurodegenerative diseases, including Alzheimer’s disease (AD). A close reciprocal relationship exists between calcium signaling and mitochondrial function. Growing evidence in a variety of AD models indicates that calcium dyshomeostasis drastically alters mitochondrial activity which, in turn, drives neurodegeneration. This review discusses the potential pathogenic mechanisms by which calcium impairs mitochondrial function in AD, focusing on the impact of calcium in endoplasmic reticulum (ER)–mitochondrial communication, mitochondrial transport, oxidative stress, and protein homeostasis. This review also summarizes recent data that highlight the need for exploring the mechanisms underlying calcium-mediated mitochondrial dysfunction while suggesting potential targets for modulating mitochondrial calcium levels to treat neurodegenerative diseases such as AD.

## 1. Introduction

With a rapidly aging population, it is expected that neurodegenerative disorders will become an increasingly pressing health issue, intensifying the urgency for effective treatment. Most neurological disorders are chronic and incurable, and their debilitating effects can continue for decades. There are approximately 50 million people living worldwide with dementia and this number is expected to reach 82 million in 2030 and 152 million by 2050 (WHO.int) [1]. Alzheimer’s disease (AD), the most common neurodegenerative disease [1], is the sixth leading cause of death [2]. In the United States alone, approximately 5 million people currently suffer from the disease, and these numbers are projected to triple by the year 2050 [2]. AD is characterized by gradual cognitive decline and memory loss, synaptic dysfunction, and neuronal death. Despite decades of research, there is no effective therapy for AD and the cause of AD remains unclear, especially due to its complex etiology. The primary histopathological hallmarks of AD are neurofibrillary tangles composed of the microtubule-associated protein tau and extracellular deposition of senile plaques composed of amyloid-beta (Abeta) peptides. Most AD research has focused on investigating the amyloid hypothesis, which proposes that aberrant Abeta production is the core cause of AD, and that all other dysfunctions observed in AD, including hyperphosphorylated tau tangles, are a consequence of Abeta neurotoxicity. Abeta plaques on their own have been shown to interfere with synaptic function and cause extensive neuronal death, especially within the cortical and hippocampal regions of the brain that are responsible for learning and memory [3]. In support of Abeta being the causative agent of AD, genetic mutations in genes encoding presenilin 1 (PSEN1), presenilin 2 (PSEN2), and the amyloid precursor protein (APP), which lead to nearly all early-onset familial AD (FAD) cases, are all involved in Abeta processing. Abeta peptides of varying lengths are generated from APP by the consecutive cleavage by beta-secretase and gamma-secretase. Mutations in the genes encoding PSEN1 and PSEN2, which underlie roughly 70% of all familial AD cases [4], form the catalytic component of the gamma-secretase complex. Increased production of the more toxic, aggregation-prone Abeta42 species relative to Abeta40, which results in the generation of extracellular amyloid fibrils and plaques, is thought to be the primary cause of AD [3]. Therapeutic strategies for treating AD have focused on reducing the burden of Abeta plaques by either preventing Abeta production or by promoting its clearance. However, gamma-secretase inhibitors have been a failure clinically, and Abeta plaque load is not strongly correlated with dementia onset or severity [5], which has prompted reconsideration of the amyloid hypothesis [6,7,8,9]. There is also considerable evidence that Abeta oligomers, as opposed to the more highly aggregated Abeta fibrils, are the primary cytotoxic species and pathological agent of AD. This soluble form has been shown to damage synapses absent Abeta plaque formation [10,11]. Additionally, oligomeric Abeta isolated from AD patients was sufficient to cause synaptic dysfunction and memory loss in mice [12]. It remains unclear whether selectively targeting Abeta oligomers will show more success in clinical trials. Indeed, the use of immunotherapies targeting Abeta, some of which have been shown to bind oligomeric Abeta with high affinity, have no effect on disease progression [8,13]. Other studies have shown that overexpression of Abeta peptides in mice, while simulating the Abeta pathology observed in AD, does not result in similar synaptic loss and memory impairment [14]. Additionally, neurodegeneration induced by presenilin mutations can occur in the absence of Abeta production [15,16]. Mutations in other components of the gamma-secretase complex (APH-1, PEN-2, and nicastrin) have not been implicated in FAD, implying APP and presenilin mutations do not cause FAD merely by altering Abeta production [17]. Furthermore, many other symptoms (e.g., increased inflammation, altered calcium signaling, mitochondrial dysfunction, oxidative damage) appear to arise independently of any Abeta involvement as they often precede Abeta plaque formation, and are also more strongly correlated with cognitive decline [18,19]. Collectively, this suggests that Abeta production is not necessarily the proximate cause of AD and that additional pathological factors must be explored to develop viable therapies.

## 2. Calcium Dysregulation in AD

The connection between calcium and AD was observed several decades ago [20], but recent data have increased support for this hypothesis, and strongly implicate a role for calcium in AD. The calcium hypothesis of AD states that disruptions to neuronal calcium signaling underlie not only amyloid plaque deposition but a series of molecular changes within the neuron that cause neuronal dysfunction [21]. During neurotransmission, a rise in intracellular calcium following membrane depolarization transmits the signal to synapses. Calcium signaling in neurons is, therefore, crucial for neurotransmission and for maintaining synaptic plasticity and generating long-term potentiation (LTP), which forms the basis of learning and memory through the progressive strengthening of synapses [22,23]. Calcium signaling also regulates neuronal metabolism and energy production which is necessary to sustain synaptic transmission [24]. Unsurprisingly, disruptions to calcium signaling have debilitating consequences on neuronal function. Evidence for calcium dysregulation in AD was initially found over 25 years ago in fibroblast cells isolated from AD patients, which showed enhanced endoplasmic reticulum (ER) calcium uptake and ER calcium release [25,26]. Further studies in AD mouse models have supported the ubiquitous involvement of calcium dysregulation in AD, linking it to memory loss and increased toxicity of Abeta peptides [27,28,29]. Other studies have shown that elevated ER or cytoplasmic calcium increases Abeta production by triggering phosphorylation of APP and tau, indicating intracellular calcium dysregulation exacerbates amyloidosis and tau pathology [30,31]. Emilsson and Jazin also showed that mRNA expression of genes involved in calcium regulation is altered in AD brains [32]. The results from this study further suggest that ER calcium channel activity is elevated in AD [32].

Many familial AD PSEN mutations are also linked to dysregulated calcium signaling [33]. In addition to PSEN1/2′s role as the catalytic subunit of gamma-secretase, PSEN1/2 regulates ER calcium stores, a function that is notably gamma-secretase independent, demonstrating that PSEN1 and PSEN2′s impact on neuronal function extends beyond Abeta generation [34,35]. PSEN1/2 are transmembrane proteins present on most endomembranes but predominate on the ER membrane, where they have been found to physically interact with ER calcium channels, including the two main ER calcium release channels, inositol 1,4,5-trisphosphate receptors (IP_3_Rs) and ryanodine receptors (RyRs) [28,36,37]. In line with this, mice carrying FAD mutations show altered activity of both IP_3_Rs and RyRs. Presenilin mutations have been documented to increase either the expression or sensitivity of RyRs and IP_3_Rs, leading to exaggerated RyR- and IP_3_R-mediated calcium release in response to various agonists [28,36,38,39,40]. Loss of PSEN1 function has also been shown in Xenopus oocytes to increase the activity of smooth endoplasmic reticulum Ca^2+^ ATPase (SERCA), which is responsible for pumping calcium into the ER to maintain cytosolic calcium levels [29]. Elevated SERCA activity in turn leads to overloading of ER calcium stores and a compensatory release of ER calcium [29]. PSEN itself may act as a passive ER calcium leak channel, which similarly results in ER calcium overfilling and exaggerated ER calcium release [35]. Notably, this elevated ER calcium release also precedes Abeta pathology and stimulates Abeta formation [29]. This is consistent with a previous study showing that altered calcium signaling in fibroblasts derived from asymptomatic FAD families was a strong predictor of future disease development [41]. Moreover, several other studies demonstrate that FAD mutations in the genes encoding PSEN1 or PSEN2 result in higher basal levels of calcium in cortical and hippocampal neurons due to excessive ER calcium release [29,36,38,39]. Elevated cytosolic calcium, in turn, impedes induction of long-term potentiation (LTP) [42]. There is also evidence that FAD mutations in APP similarly cause increased intracellular calcium concentrations independent of Abeta involvement [43,44]. Abeta itself has also been shown to derive some of its cytotoxic effects from promoting cytosolic calcium influx [45]. Other studies in cultured cortical neurons showed that oligomeric Abeta promoted ER calcium release, which led to cell death [46]. Altogether, the data strongly suggest that exaggerated ER calcium release and disruptions to calcium homeostasis play an important role in AD pathogenesis.

## 3. Mitochondria and ER Crosstalk in Presenilin Mutants, and in Sporadic AD

Both ER and mitochondria are dynamic organelles that are actively moving within the cell and make transient contacts to facilitate crosstalk between the two organelles [47]. The mitochondria-associated membranes (MAMs) are regions where the ER closely associates with the outer mitochondrial membrane (OMM). These MAMs contain a specialized distribution of phospholipids and proteins to enable ER–mitochondria communication [48]. MAMs regulate a variety of processes including mitochondrial metabolism and energy production, the ER stress response, lipid synthesis, and apoptosis signaling [48]. Calcium homeostasis is also highly dependent on MAM function (Figure 1). IP_3_R and RyR localization is concentrated at the MAMs to promote the rapid uptake of calcium into the mitochondria [49] (Figure 1). MAMs are also enriched in voltage-dependent anion channels (VDACs), an ion channel on the OMM that regulates the transportation of a variety of ions and metabolites into and out of the mitochondria, and is primarily responsible for calcium uptake into the mitochondria across the OMM [50,51] (Figure 1). There is growing evidence that ER–mitochondrial communication is perturbed in AD. It has been shown in neurons of both sporadic and familial AD patients and an AD mouse model that there are increased ER–mitochondria contact points and expression of MAM-associated proteins, including IP_3_Rs, RyRs, and VDACs [50]. Abeta exposure to hippocampal neurons was also shown to increase ER–mitochondrial contact and promote transfer of calcium from the ER into the mitochondria [50]. Similarly, cells expressing the ε4 allele of apolipoprotein E (APOE4), which is considered a major risk factor for developing sporadic AD [52], show upregulated MAM activity and ER–mitochondrial communication [53], further suggesting disruption of MAM function is a common characteristic observed in AD. PSEN1/2 localization is also concentrated at the MAMs [54] (Figure 1). Accordingly, PSEN1 and PSEN2 FAD mutations have been shown to alter lipid and phospholipid synthesis, lipid exchange, and calcium transfer between the ER and mitochondria [55,56]. Moreover, presenilin and APP FAD mutations have been shown to increase the number of ER–mitochondria contact sites [57]. In *Caenorhabditis elegans*, which has a highly conserved presenilin ortholog but does not produce Abeta peptides [58], presenilin loss promotes ER-to-mitochondria calcium uptake, suggesting that presenilin alters ER–mitochondria calcium transfer via an Abeta-independent mechanism [59]. This study further demonstrated that the function of presenilin in mediating ER–mitochondrial calcium signaling is independent of gamma-secretase activity. However, there is also evidence that changes to MAM function or connectivity may alter or increase gamma-secretase activity due to presenilin enrichment at the MAM, and this may in turn promote pathologic Abeta42 generation [60]. Considering MAMs are involved in processes that are frequently disrupted in AD, including calcium homeostasis, it is likely the altered MAM distribution and functions are involved in AD pathogenesis. The increased ER–mitochondrial communication observed across multiple AD models has especially important implications on mitochondrial function. Indubitably, neurons are highly sensitive to mitochondrial defects and alterations in mitochondrial oxidative respiration.

## 4. Mitochondrial Calcium and AD

Neurons are particularly reliant on mitochondria for energy generation to sustain synaptic transmission. The brain accounts for only 2% of human body weight, but uses roughly 20% of the body’s oxygen supply [61], The vast majority of neuronal ATP is produced through mitochondrial oxidative respiration [62,63], and neurons use this ATP primarily to generate the ionic gradients necessary for synaptic transmission [64]. Mitochondria are highly mobile organelles, and their subcellular localization impacts their ability to provide ATP to various cellular compartments [65]. In neurons, mitochondria are predominantly localized to the synapses to meet the energy demand at these sites [64]. Preserving mitochondrial quality and the mitochondrial trafficking network is thus especially important for neurons, as both are required to maintain synaptic plasticity and ultimately the learning and memory process [66,67]. Consequently, neurons are highly sensitive to mitochondrial defects and alterations in mitochondrial oxidative respiration [64]. Mitochondrial dysfunction is a common feature across neurodegenerative diseases, including AD [66,67]. In fact, mitochondrial dysfunction is thought to be one of the main drivers of the disease [68,69].

Intracellular calcium greatly impacts mitochondrial function. Indeed, calcium plays a direct role in stimulating enzymes of the tricarboxylic acid (TCA) cycle and electron transport chain leading to increased oxidative phosphorylation [70,71,72,73] (Figure 1). Mitochondria, in turn, regulate cellular calcium signaling by sequestering and buffering cytosolic calcium. As mentioned, the positioning of the ER calcium channels at the MAMs facilitates calcium transfer into the mitochondria. Selective transport of calcium into the matrix across the inner mitochondrial membrane (IMM) is accomplished by the highly calcium selective mitochondrial calcium uniporter (MCU) protein complex [74,75,76] (Figure 1). The MCU complex is composed of four core components: the pore forming MCU protein, an auxiliary subunit EMRE (essential for MCU regulator) and the MICU gatekeepers, MICU1 and MICU2/3 [77,78,79]. The MCU complex regulates calcium uptake into the matrix primarily through the MICU1 and MICU2/3 proteins that sense calcium through their conserved calcium-binding EF hand domains [80]. MICU1 and MICU2 are widely expressed in most mammalian tissues, whereas MICU3 is expressed only in skeletal muscles and the CNS [81]. MICU2 and MICU3 have similar structure and function in regulating MCU complex activity [82,83]. Recently, MICU2 has been shown bind to MICU1 and together these proteins allow for gatekeeper activity. Specifically, elevation in cytosolic calcium promotes calcium binding to the EF hands of the MICU1-MICU2 heterodimer, enabling MICU1 to facilitate MCU activity allowing calcium entry into the mitochondria [84,85]. MICU1 activation is also influenced by the activity and expression of cytosolic calcium binding proteins, which also have EF-hand domains that compete with MICU1 for calcium binding [86]. By promoting calcium uptake when cytosolic levels are high, mitochondria buffer calcium to maintain intracellular calcium homeostasis. This uptake is energetically favorable due to the negative membrane potential generated by transport of H^+^ across the IMM by the electron transport chain, making mitochondria well suited for this task. The mitochondria at synapses are important not only for ATP delivery but also for tightly regulating calcium concentration at the synapses for effective neurotransmission [87]. Considering the abundant evidence implicating intracellular calcium dysregulation in AD, it is likely that regulation of intracellular calcium through mitochondrial calcium buffering is a factor in this process.

Calcium signaling also plays a direct role in mitochondrial activity [88]. Mitochondrial calcium levels significantly impact mitochondrial activity and cellular ATP supply. Calcium uptake into the mitochondria increases oxidative phosphorylation by stimulating the activity of the F_1_F_0_-ATP synthase and enzymes within the TCA cycle, specifically alpha-ketoglutarate dehydrogenase, isocitrate dehydrogenase, and pyruvate dehydrogenase [88]. Mitochondrial activity is thus highly sensitive to calcium levels. Evidence suggests that along with increased ER–mitochondrial communication and elevated cytosolic calcium, mitochondrial calcium is elevated in AD. Due to MICU1 and MICU2/3 calcium-sensing and gating properties, increased cytosolic calcium can promote mitochondrial calcium uptake [74]. Multiple studies have also reported that expression of cytosolic calcium binding proteins calmodulin, calbindin D28K, and parvalbumin is reduced in AD patients and AD models, which would presumably free up calcium to bind MICU1 and MICU2/3 and activate MCU [89,90,91,92]. Therefore, it is unsurprising that mitochondrial calcium is elevated in AD models. Moreover, Abeta exposure has been shown to increase mitochondrial calcium levels in cortical neurons that promotes neurodegeneration, which can be reversed by blocking MCU [50]. There is evidence that the oligomeric form of Abeta produces calcium-permeable pores in the mitochondrial membrane, which promotes calcium uptake, suggesting that Abeta’s toxicity may result in part from its ability to disrupt mitochondrial calcium homeostasis [93,94]. Similarly, an additional study demonstrated that Abeta oligomers increased mitochondrial calcium levels by promoting ER calcium release, resulting in mitochondrial dysfunction [95]. Furthermore, a separate study looking at a mouse model with FAD mutations in PSEN1 and APP showed that these animals had elevated mitochondrial calcium [96]. Mitochondrial calcium levels are also increased in *C. elegans* presenilin mutants, which can be abrogated by preventing ER calcium release or mitochondrial calcium uptake, suggesting a conserved role for presenilin in maintaining mitochondrial calcium homeostasis by regulating ER calcium release at the MAMs [59]. Importantly, reduction of mitochondrial calcium levels in the *C. elegans* presenilin mutants restores neuronal function.

What are the consequences of elevated mitochondrial calcium? Although mitochondrial calcium homeostasis can be restored through calcium efflux pathways, which are regulated primarily through the sodium–calcium exchanger NCLX [97], excessive calcium uptake or impairments to calcium efflux can overwhelm mitochondrial calcium capacity. Mitochondrial calcium overload, which when combined with other stressors such as oxidative damage, results in the formation and opening of the mitochondrial permeability transition pore (mPTP) [98] (Figure 1). Although the molecular components of the mPTP are still under debate, the activity of the mPTP is known to span both the OMM and IMM, and its opening induces calcium efflux from the matrix. However, prolonged opening of the mPTP leaves the mitochondria open to the osmotic influx of cytosolic solutes and water, which causes the matrix to swell and rupture. Cytochrome c is also released from the mitochondria from prolonged mPTP opening, leading to the initiation of apoptosis. This process has been observed in several AD mouse models. In the aforementioned study showing elevated mitochondrial calcium in AD mice, high mitochondrial calcium correlated with the induction of apoptosis, whereas neurons containing mitochondria with normal calcium concentrations did not undergo apoptosis [96]. Another study showed that Abeta42 exposure could cause neuronal apoptosis by promoting mitochondrial calcium overload and stimulating the opening of the mPTP [99]. Excessive mitochondrial uptake in such contexts would present a particular problem for aged mitochondria, which are particularly vulnerable to high calcium levels, as they have lower calcium buffering capacities and are more susceptible to calcium overload [100].

## 5. Calcium-Induced Changes to Mitochondrial Activity Promote ROS Production

Elevated mitochondrial calcium has also been shown to disrupt neuronal function through the elevated production of reactive oxygen species (ROS) [101]. Mitochondrial oxidative respiration is a major source of ROS, as ROS are produced as a byproduct of the reduction of oxygen [102]. The energy used to pump H^+^ across the mitochondria’s inner membrane space is generated by a series of electron transfer reactions along the electron transport chain where each electron transport chain complex has increasingly greater reduction potential, the final electron acceptor being oxygen. Although electrons transferred sequentially along the electron transport chain will react at the end with oxygen to produce water, electrons may also be passed to oxygen prematurely, resulting in partial reduction of oxygen to superoxide anion (O_2_^•−^). This superoxide radical is a major source of oxidants and free radicals in the cell, as it reacts to produce hydrogen peroxide, which again reacts to produce the highly toxic hydroxyl radical (·OH). Although ROS at low levels provide important cellular functions by acting as signaling molecules, excessive ROS are toxic to the cell. As a site of superoxide generation, mitochondria are exposed to an especially high level of ROS, which can damage mitochondrial DNA and proteins over time, leading to severe mitochondrial damage and inefficiency. Dysfunctional mitochondria, in turn, generate greater levels of ROS; in this way, oxidative stress and mitochondrial dysfunction are closely linked. It has long been theorized that the aging process is caused in part by the progressive accumulation of ROS-induced cellular damage, both as a result of increased ROS generated by an inefficient electron transport chain or failure of antioxidant systems [103]. If the collective damage to proteins, DNA, and organelles induced by ROS is too great, the cell will undergo apoptosis.

Like mitochondrial dysfunction, oxidative stress plays a significant role in the pathogenesis of AD [104,105]. In vivo and in vitro studies show a direct relationship between oxidative stress and AD [105,106,107]. Furthermore, elevated ROS is highly correlated with the early stages of AD [108] and precedes Abeta plaque formation [109]. Oxidative stress can also promote tau hyperphosphorylation and fibril formation [110]. In a variety of AD animal models, high ROS levels have been shown to cause extensive neuronal death and cognitive decline [111]. Neurons are especially vulnerable to oxidative damage induced by mitochondrial respiration due to their especially high energy and oxygen demand, which in turn produces greater relative levels of ROS [98]. Their high lipid content also makes them vulnerable to lipid peroxidation by ·OH [98]. It has also been proposed that the regions of the brain that first undergo neurodegeneration in AD occur due to their increased susceptibility to oxidative damage, which can result from greater energy demands or higher basal levels of ROS required for signaling [112]. Elevated mitochondrial calcium levels have been associated with increased oxidative stress in AD models. Impairment to NCLX function that prevents mitochondrial calcium efflux in a mouse AD model resulted in excessive mitochondrial calcium and oxidative stress, which in turn led to amyloid and tau pathology and ultimately neuronal death [113]. Rescue of NCLX function was able to prevent cognitive defects in these animals. Buildup of mitochondrial ROS as a result of mitochondrial calcium overload also triggered apoptosis in a separate AD mouse model [99]. Mitochondrial calcium influx has also been shown to induce mitochondrial damage by stimulating oxidative phosphorylation, increasing the amount of ROS generated as a byproduct (Figure 1). Abeta oligomers also damage neurons by elevating lipid peroxidation and oxidative stress, further supporting the idea that oxidative damage underlies AD pathogenesis [114,115,116]. High mitochondrial calcium, fragmentation of mitochondria, oxidative stress and neuronal death have also been observed in APP/PS1 transgenic mice and in the brains of AD patients prior to Abeta formation [33,99,117]. Fibroblasts isolated from FAD patients showed both elevated oxidative respiration and ROS, which could be blocked either by preventing mitochondrial calcium uptake or by reducing respiration [94]. Similarly, in astrocytes differentiated from iPSCs derived from FAD patients bearing PSEN1 mutations, the mutant astrocytes showed higher oxygen consumption that resulted in greater ROS levels [118]. Notably, this alteration in metabolism was gamma-secretase independent. In *C. elegans*, neuronal and behavior defects resulting from presenilin dysfunction was dependent on elevated mitochondrial calcium-induced oxidative respiration and concomitant ROS generation [119]. Indeed, by limiting ER calcium release or mitochondrial calcium uptake in the *C. elegans* presenilin mutants reduced mitochondrial generated ROS and neurodegeneration. Furthermore, the neuronal defects observed in the *C. elegans* presenilin mutants were suppressed by treating these animals with the mitochondrial directed antioxidant, MitoTEMPO [94]. These studies indicate that aberrant mitochondrial activity induced by altered calcium signaling may be a mechanism by which FAD mutations generate ROS and lead to neuronal dysfunction. Therefore, this initial mitochondrial hyperactivity caused by aberrant mitochondrial calcium influx may in turn accelerate mitochondrial impairment, resulting in reduced ATP generation insufficient for neuronal synaptic transmission. Taken together, these data indicate that altered mitochondrial calcium homeostasis in FAD models leads to mitochondrial dysfunction and ROS production that promotes neurodegeneration.

## 6. Mitochondrial Function in Protein Homeostasis

The preservation of a highly functional proteome is vital for the aging nervous system. Indeed, protein misfolding and aggregation is a common manifestation observed in neurodegenerative diseases [120,121]. Therefore, the integrity of the protein homeostasis (proteostasis) network is critical for cell function and survival and is particularly important for non-dividing cells such as the neurons of the central nervous system. Insults that affect proteostasis include oxidative stress [122]. Accordingly, perturbations in mitochondrial function resulting in elevated production of ROS will greatly impact the maintenance of the proteome. Proteins that are irreversibly oxidized need to be removed by the proteostasis network; however, as organismal aging occurs, the efficiency of the proteostasis network diminishes, leading to protein misfolding and aggregation. In *C. elegans*, a clear link between altered calcium uptake into the mitochondria and the impact this has on proteostasis has been demonstrated. Mutations in the gene encoding presenilin in *C. elegans* result in elevated ER to mitochondrial calcium transfer. The elevation in mitochondrial calcium results in mitochondrial hyperactivity, namely increased oxidative phosphorylation and ROS production that promotes neuronal dysfunction [119]. The elevated oxidative stress caused by increased mitochondrial calcium leads to proteostasis defects in the presenilin mutants that can be rescued by inhibiting mitochondrial calcium uptake or by treating the animals with antioxidants [123] (Figure 1). The two main pathways involved in proteome maintenance are the ubiquitin-proteasome system and the autophagy-lysosome system. While the ubiquitin-proteasome activity appeared normal in the *C. elegans* presenilin mutants, there was a clear reduction in the formation of autophagosomes suggesting a defect in the autophagy-lysosome system [123]. Findings from FAD mouse and cell culture models also demonstrate that calcium dysregulation is linked to defects in autophagy [124]. Interestingly, the nutrient and energy sensing mechanistic target of rapamycin complex 1 (mTORC1) signaling pathway is a central inhibitor of autophagy and has been reported to be upregulated in AD as well as other neurodegenerative disorders [125,126,127,128,129]. However, the mechanism underlying the upregulation of mTORC1 signaling in these disorders is not clear. Nevertheless, these studies highlight the importance of mitochondrial calcium homeostasis and the impact it can have on proteostasis.

## 7. Mitochondrial Calcium-Dependent Subcellular Localization and Transport

As mentioned, trafficking and localization of mitochondria is essential for neuronal function. Mitochondria are extremely dynamic organelles undergoing constant fission and fusion events and they migrate along microtubule tracks to ensure their localization throughout the neuron. Principally, the fission and fusion events are regulated by three large GTPases, namely mitofusins (Mfn1/2), optic atrophy1 (OPA1) and dynamin related protein 1 (Drp1) and the trafficking along microtubules is mediated by the kinesin and dynein motors through the action of the adaptor protein Milton/Trak and the atypical Rho GTPase Miro [130,131,132]. These pathways are interconnected and are required to balance mitochondrial shape, morphology and function. Given the critical role of mitochondria in neurons, impaired mitochondrial dynamics is increasingly implicated in neurodegenerative diseases [133,134,135]. Additionally, mitochondrial fission and fusion dynamics have been associated in neurotransmitter release and terminal axon branching by regulating cytosolic calcium levels [136]. Trafficking of mitochondria is regulated in part by calcium and is mediated via Miro 1 and Miro 2 [137]. Miro proteins are transmembrane proteins that are localized to the OMM and contain two-conserved EF-hand calcium binding domains flanked by two Rho-like GTPase domains [133,138,139]. Miro interacts with Milton/Trak to link mitochondria to kinesin and dynein to allow for trafficking along the microtubule cytoskeleton [140,141]. When cytosolic calcium levels increase, calcium binds to the Miro EF-hands creating a conformational change that disengages Miro and the mitochondria from the microtubule network [132,142]. Consequently, an increase in calcium can pause the trafficking of mitochondria along the microtubules. It has also been demonstrated that Miro can influence mitochondrial trafficking by directly mediating mitochondrial calcium uptake via the MCU complex. In this study, the authors found that increased mitochondrial matrix calcium influx inversely correlated with mitochondrial trafficking speed [143]. Thus, cytosolic as well as mitochondrial calcium levels, can influence the localization and transport of mitochondria. In addition to mediating mitochondrial trafficking and mitochondrial calcium uptake, Miro proteins have also been implicated in several other mitochondrial functions, including IMM and OMM organization, mitochondrial fission, ER–mitochondrial contacts, and mitophagy. In both Drosophila and mammalian cells, Miro was shown to promote ER–mitochondrial interactions [144,145]. Moreover, in Drosophila, Miro was shown to facilitate ER to mitochondria calcium signaling [144,146]. In *C. elegans* and mammalian cells, Miro was shown to mediate mitochondrial fission upon an increase in cytosolic calcium that is independent of the conical Drp1 fission machinery [147,148]. Notably in mammalian cells, this fission event facilitates the turnover of mitochondria through the process of mitophagy, which is the selective removal of defective mitochondria by autophagy [147]. Furthermore, another study using human induced pluripotent stem cell-derived neurons from Parkinson’s disease patients found that targeted loss of Miro enables the removal of damaged mitochondria by mitophagy and promotes neuronal fitness [149]. It is unclear whether the trafficking function of Miro is required to mediate these additional activities of Miro (e.g., ER–mitochondria contact, fission or mitophagy). Nonetheless, these data highlight the pleiotropic role Miro and calcium have on mediating mitochondrial localization and function.

## 8. Conclusions

The influx of calcium into the mitochondria has long been established as a key regulator of many cellular homeostatic processes that range from energy production to cell death and necrosis [150,151]. The recent molecular discovery of the MCU complex components has provided critical insight into the role mitochondrial calcium influx has in energy production under increased work load but also, paradoxically, in promoting disease, such as neurodegeneration [152,153]. For example, studies in mice and *C. elegans* AD models have shown that there is an increase in mitochondrial calcium levels that can be reduced by blocking the MCU complex [99,119]. Critically, it was demonstrated in the mouse AD model that the high levels of mitochondrial calcium precede neuronal dysfunction and that this dysfunction can be rescued by inhibiting the MCU complex and blocking mitochondrial calcium uptake [99]. Similar observations have been made studying other neurodegenerative diseases [154,155,156]. For instance, recent work in zebrafish and Drosophila models of Parkinson’s disease have discovered elevated mitochondrial calcium levels that when reduced, could alleviate neurodegeneration [146,157]. These data highlight the significance mitochondrial calcium homeostasis has in neurodegeneration and point to the importance of understanding the mechanisms mediating mitochondrial calcium influx and activity. Indeed several recent review articles have suggested that targeting the MCU complex could be a potential therapeutic target for treating neurodegenerative diseases [158,159,160].

## Figures and Tables

**Figure 1 ijms-21-09153-f001:**
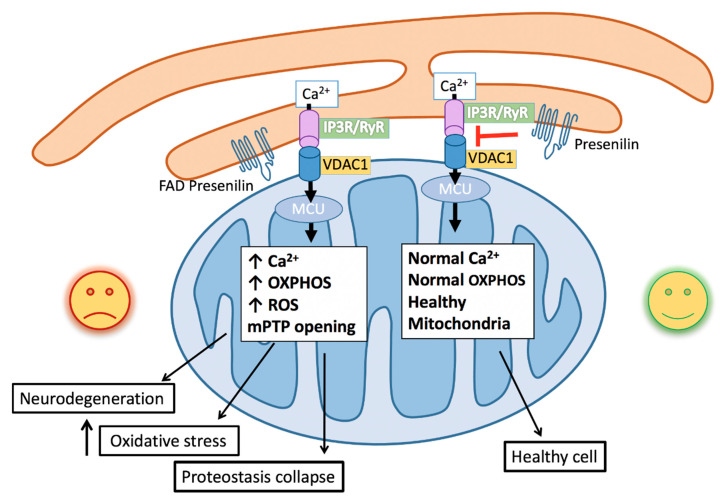
Schematic of familial Alzheimer’s Disease (FAD) mutations facilitating endoplasmic reticulum (ER)–mitochondrial calcium transfer. Increased calcium release from the IP_3_ receptor (IP3R) and Ryanodine receptors (RyR) at the ER mitochondrial-associated membranes (MAMs) is taken up through the voltage-dependent ion channel (VDAC) on the mitochondrial outer membrane and the calcium-selective mitochondrial calcium uniporter (MCU) on the mitochondrial inner membrane. Increased mitochondrial calcium stimulates oxidative phosphorylation, leading to increased reactive oxygen species (ROS) generation, which promotes oxidative stress, mitochondrial permeable transition pore (mPTP) opening and apoptosis, protein misfolding and proteostatic collapse, and neurodegeneration.

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
