# Peer review of "The Role of Mitochondrial Calcium Homeostasis in Alzheimer’s and Related Diseases"

_ijms, 2020, doi:10.3390/ijms21239153_

Round 1

Reviewer 1 Report

The article summarizes changes in calcium homeostasis in Alzheimer's disease, focusing on mitochondrial calcium. The work is well written and structured, focusing the main topics on the subject. The work is suitable for publication, but there are two aspects that must be taken into account: - the role of Abeta oligomers in synaptic dysfunction and thus in early cognitive disorders is not discussed, in addition, no reference is made to studies showing the effect of oligomeric Abeta on calcium homeostasis, namely on mitochondrial calcium (eg 10.1016 / j .neurobiolaging.2014.09.006) - reference is not also made to the effect of the Abeta peptide on the content and release of Ca2+ from the ER (e.g. 10.1016 / j.nbd.2006.05.011). Finally, the seminal studies of P Pizzo supporting the role of AD-associated mutations on Ca2 + homeostasis should be discussed. (e.g. 10.1080 / 15548627.2019.1596489; 10.1007 / s40520-019-01341-0.)                

Author Response

We greatly appreciate the comments and suggestions made by the reviewer. While the review was positive, the reviewer was concerned about the lack of attention discussing 1) the role of Abeta oligomers in synaptic dysfunction, mitochondrial calcium homeostasis and ER calcium release and 2) P. Pizzo’s findings that support a critical role of FAD mutation in calcium homeostasis. We have addressed this lack of attention by adding discussion addressing the role of Abeta oligomers in synaptic dysfunction on lines 58-59, mitochondrial calcium homeostasis on lines 212-215, and ER calcium release on lines 215-217. Lastly, we have addressed the lack of citing P. Pizzo’s work by discussing their critical discoveries on lines 147-148, 325-326.

Reviewer 2 Report

This is a timely and original review focused on mitochondrial calcium alterations in Alzheimer’s disease. It is written concisely and follows a logical outline. The authors provide clear arguments supported by good standing research publications to support the major hypothesis of this review, and the points addressed have been clearly explained. I only have minor corrections and suggestions for this review (see below). Therefore, I recommend to publish this review with minor corrections.

Minor corrections and suggestions:

  1. The hydroxyl radical should be written correctly in this manuscript (lines 229 and 247), namely ·OH instead of the hydroxyl anion chemical formula OH-.
  2. The expression “the molecular identity of the mPTP is not known” (lines 204-205) is a too strong assertion as is. I suggest to change to something like “the molecular components of the mPTP and their modulation by ROS are still under debate”.
  3. I miss a mention to the cellular (or neuronal) toxicity of amyloid beta peptides oligomers, not only mention Abeta plaques toxicity in the section 1, as Abeta oligomers have been shown to be the more toxic Abeta species for neurons.
  4. I also suggest to briefly mention the pro-oxidant properties of amyloid beta peptides monomer and oligomeric states, owing to the putative relevance for the content of section 5.
  5. Owing to the relevance of reaching a cytosolic calcium threshold for MCU complex activation, lines 171-172, a brief comment on the reported changes of cytosolic calcium buffering proteins in AD (calmodulin, parvalbumin and calbindin D28K). Note also that these proteins will compete for calcium binding to other proteins having EF-hand domains cited in this review, like MICU (section 4, lines 169-172) and Miro-proteins (lines 325-326). 

Author Response

Reviewer 2

We greatly appreciate the comments and suggestions made by the reviewer. Although the review was overly positive, the reviewer had some minor suggestions and corrections for the manuscript. These include:

  • “The hydroxyl radical should be written correctly…”. We agree and thank the reveiwer for catching this mistake and the correction have been made (lines 254, 272)
  • “The expression “the molecular identity of the mPTP is not known” is too strong an assertion as is. I suggest to change to something like “the molecular components of the mPTP and their modulation by ROS are still under debate”. We have changed the language to “Although the molecular components of the mPTP are still under debate, …” on line 230.
  • No mention of cellular toxicity of amyloid beta in other sections besides section 1. We have added discussion of Abeta in other sections (Section 2: lines 88-92, 122-123, Section 3: 141-142, Section 4: lines 210-217, 238-240, Section 5: lines 283-285.)
  • The reviewer also suggested to briefly mention the pro-oxidant properties of Abeta. This is now included on lines 283-285.
  • The reviewer suggested to add a comment addressing the role of calcium buffering proteins in AD. We have added details addressing other calcium buffering proteins on lines 189-191, 205-210.